# Construction of Overexpression Vector with *TYR7523* Gene and Its Effect on Browning in *Macrocybe gigantea*

**DOI:** 10.3390/jof11030216

**Published:** 2025-03-12

**Authors:** Jinyun Gao, Shuqing Song, Xinqian Liu, Zhuanlin Mo, Meihua Mo

**Affiliations:** 1College of Food Science, South China Agricultural University, Guangzhou 510642, China; 18948298617@163.com (J.G.);; 2Center for Basic Experiment and Practical Training, South China Agricultural University, Guangzhou 510642, China

**Keywords:** *Macrocybe gigantea*, tyrosinase gene, overexpression, browning

## Abstract

*Macrocybe gigantea* is a rare high-temperature edible fungus known for its resistance to browning. Previous studies suggested that the anti-browning property of the SCAU4 strain might be associated with low expression levels of the *TYR7523* gene. In this study, an overexpression vector for the *TYR7523* gene was constructed and introduced into SCAU4 mycelium using an Agrobacterium-mediated transformation method. After three rounds of hygromycin resistance screening, successful transformants were identified through PCR amplification and validated by qRT-PCR analysis, confirming a 3.47-fold upregulation of *TYR7523* expression. The overexpression strain *OE7523* was compared with the wild-type SCAU4 strain in terms of growth rate, browning degree, and tyrosinase activity. Although there was no significant difference in growth rate on the mother culture medium, *OE7523* showed faster growth on the stock culture and mycelium culture medium. In the late storage period, *OE7523* exhibited a higher browning degree and tyrosinase activity than SCAU4, suggesting a potential role of *TYR7523* in fruiting body browning. Physiological analyses indicated that low *TYR7523* expression may contribute to storage tolerance, while high expression influenced postharvest browning and preservation duration. The results provide data support for further study on the function of *TYR7523* gene of *Macrocybe gigantea*.

## 1. Introduction

*Macrocybe gigantea* (Massee) Pegler & Lodge belongs to the Basidiomycota, Agaricomycetes, Agaricales, and Tricholomataceae. *Macrocybe* is a rare high-temperature edible mushroom [1,2]. Its fruiting body has a delicious taste and high medicinal value [3,4,5], containing abundant crude protein, crude fiber, polysaccharides, and various mineral elements [6,7,8,9]. Compared with the common mushroom bisporus in the market, the *Macrocybe gigantea* has the storage and transportation resistance required for commercialization. The fruiting body is preserved for 30 d at 8~12 °C. Therefore, the study of its anti-browning and storage resistance mechanism is of great significance for the storage and preservation of mushrooms. Yu X.M., et al. [10,11] found storage period of huge mushroom laccase activity is far higher than mushroom bisporus, and grass mushroom, and compared to brown speed is much slower, speculated that laccase is not caused by huge mushroom brown key enzyme, at the same time huge mushroom polyphenols oxidase and tyrosinase activity is declining, speculated its storage and transportation characteristics and polyphenol oxidase and tyrosinase activity is low.

Browning is an important factor that causes the decline of quality, storage, and transportation performance of edible fungi and is an important link in the commercial development of edible fungi [12,13,14]. Tyrosinase genes are involved in many physiological responses, such as mammal skin pigmentation, browning of fruits and vegetables in plants, defense systems in arthropods, and differentiation of reproductive organs and spore formation in fungi [15,16,17]. Excessive accumulation of melanin is not only responsible for the browning of fruits and vegetables but also for disordered pigmentation. The vast majority of current studies on tyrosase inhibition of tyrosinase, develop melanin inhibitors through downregulation of tyrosinase.

In the early stage of this study [18,19], the whole genome of the *Macrocybe gigantea* was used as the reference to annotate the transcripts of the *Macrocybe gigantea* treated with ferrous ion stress and the transcripts without treatment and obtained three tyrosinase gene sequences: *TYR7523*, *TYR158*, and *TYR6843*. At the same time, qRT-PCR verified that the expression of *TYR7523*, a key factor of the mushroom, was significantly up-regulated under iron stress, which confirmed that the mushroom was not easy to brown due to the low expression of tyrosinase. Wu Ying [2] combined transcriptome sequencing to analyze the physiological and biochemical indicators of eight developmental stages, and the results showed that TYR activity was relatively constant and the trend was wavy. The differential expression of genes in the metabolic pathways related to melanin production is mainly enriched in phenylalanine metabolism, tyrosine metabolism, ubiquinone, and other terpenoid biosynthesis and ketone bodies, so it is speculated that the differential expression of genes may be directly related to the degree of browning and storage quality of different developmental fruiting bodies during storage [20,21].

In this study, the function of tyrosinase gene *TYR7523* in SCAU 4 strain, Is beneficial to deeply understand the regulatory mechanism of its browning and its fruiting body development, To clarify the regulatory effect of tyrosinase on the browning and development of giant mushrooms and the specific process of its participation; Meanwhile, the constructed *TYR7523* overexpression vector was introduced into the SCAU 4 strain, By subculture as well as the mushroom production experiment, Obtained the giant mushroom overexpressing *OE7523* strain, To further explore the regulatory effect of *TYR7523* on Macrocybe gigantea, In order to improve the breeding of *Macrocybe gigantea* and other edible fungi, Accelerate the development and utilization of *Macrocybe gigantea* and other edible fungus resources to provide theoretical guidance, To lay a solid foundation for the later study of other gene functions, It provides new ideas for the mechanism and breeding of edible fungi.

## 2. Materials and Methods

### 2.1. Materials

#### 2.1.1. Test Strain

The SCAU4 strain of *M. gigantea* was isolated and preserved in Laboratory 505 of the College of Food Science, South China Agricultural University, and its fruiting bodies were cultivated in the Edible Fungi Base of South China Agricultural University. The dual carrier plasmid4 ATMT was gifted by Professor Xie Baogui from Fujian Agriculture and Forestry University. Both DH5 α *E. coli* competent cells and EHA105 Agrobacterium competent cells were purchased from Beijing Qingke Biotechnology Co., Ltd. (Beijing, China).

#### 2.1.2. Reagents and Primers

*Apa Ι*, *Sac Ι*, pMD18-T, and PrimeScript^TM^ II 1st Strand cDNA Synthesis Kit were purchased from Takara Co. (Kyoto, Japan); Plasmid pCAMBIA1303-TrpC-Hygro-gpdA-GFP purchased from Wuhan Miaoling Biotechnology Co., Ltd. (Wuhan, China); Ezup Column Fungi Genomic DNA Purification Kit was purchased from Biotech Co., Ltd. (Beijing, China); Trelief^®^ Plasmid Mini Kit Plus and 2×TSINGKE^®^ Master qPCR Mix (SYBR Green I) was purchased from Beijing Qingke Biotechnology Co., Ltd. (Beijing, China); ClonExpress MultiS One Step Cloning Kit purchased from Nanjing Novozan Biotechnology Co., Ltd. (Nanjing, China). Primers were synthesized by Beijing Liuhe Huada Gene Technology Co., Ltd. (Beijing, China).

#### 2.1.3. Culture Medium

Potato glucose agar medium (PDA), LB agar medium, IM Induction Medium Formula, Co-culture Medium Formula; PDA Screening Medium Formula (Reference: Liyan Cha [22,23], *Macrocybe gigantea* Primary Culture Medium Formula [24].

### 2.2. Cloning of gpd Promoter and TYR7523 Gene

The *M. gigantea* SCAU4 mycelium was inoculated onto a regular PDA plate and cultured at 30 °C until the mycelium grew to the full extent on the plate. The mycelium was scraped and ground in liquid nitrogen. The DNA of the *M. gigantea* was extracted by using a fungal DNA extraction kit. The total RNA of the *M. gigantea* was extracted by using the Trizol method and reverse transcribed into cDNA. Using DNA and cDNA as templates, the gpd promoter and *TYR7523* gene of the giant mushroom were amplified (Table 1). In order to ensure the correctness of the obtained target fragment, the recovered fragment was ligated with pMD18-T vector and transferred into DH5 α *E. coli* competent cells by heat shock method. Positive clones in the bacterial liquid were detected by PCR, and then they were sent to the biotechnology company for sequencing.

### 2.3. Construction of TYR7523 Gene Overexpression Vector

Primers for inserting fragments were designed using CE Design V1.04 software (Table 1), and PCR amplification was performed using plasmids pMD gpd, pMD *TYR7523*, and pCAMBIA1303-TrpC Hygro-gpdA GFP as templates. Then, plasmids plasmid 4 ATMT were double enzyme digested with Apa Ι and Sac Ι to obtain linear vectors. The purified gpd, *TYR7523*, linear vector plasmid 4-ATMT, and NOS terminator were mixed in a certain proportion, and they were connected according to the steps of homologous recombination ClonExpress MultiS One Step Cloning Kit (From Nanjing Novzan Biotechnology Company, Nanjing, China). The recombinant plasmid was transferred into DH5 α *E. coli* competent cells, and the *TYR7523* overexpression vector *plastid4-GT7523* was obtained after validation by bacterial liquid PCR, enzyme digestion, and sequencing.

### 2.4. Agrobacterium-Mediated Genetic Transformation

The vector *plasmid4-GT7523* was transferred into the competent cells of EHA105 of Agrobacterium tumefaciens by using the freeze–thaw method. The concentrations of 20 μg/mL Rif and 50 μg/mL Kan were used to screen for positive transformants. Single colonies were selected and shaken, and 1 μL of bacterial solution was taken for PCR amplification to verify whether the vector was transferred into *A. tumefaciens*. The Agrobacterium culture medium containing overexpression vectors that have been verified was expanded. The bacterial cells were collected by centrifuging at room temperature for 10 min, and resuspending the bacterial cells in IM liquid medium (containing 200 μM AS) until the OD600 was around 0.5. The mycelium of *M. gigantea* SCAU4 was taked and co-cultured with the pre-cultured IM bacterial solution for 30 min. Then it was transferred into an IM solid plate (containing 200 μM AS) and cultured upright at 25 °C. After 36 h, it was transferred to PDA screening medium containing 70 μg/mL Hyg and 300 μg/mL Cef for culture.

### 2.5. Identification of Transformants

The transformants that can be passaged three times on PDA screening medium were selected and inoculated onto regular PDA medium. After 7 days of cultivation at 30 °C, the mycelium was collected and the DNA of each transformant was extracted. The hygromycin validation was conducted by Using universal primers Hyg-f and Hyg-r.

The positive transformant and the starting strain SCAU4 were inoculated on PDA plates, cultured at 30 °C for 7 days, and the mycelium was collected. The total RNA of the positive transformant and the starting strain SCAU4 were extracted and reverse-transcribed into cDNA. Using β—tubulin as the internal reference gene, the relative expression level of *TYR7523* gene in the positive transformant and starting strain SCAU4 were detected by qRT PCR. Three replicates were performed for each sample, and the results were calculated using the 2^−ΔΔCT^ method.

The cultivated seeds containing *plasmid4-GT7523* transformants and the starting strains were cased soil for mushroom cultivation. The fruiting body of the overexpression strain and the starting strain were randomly picked for tissue isolation and cultured at 30 °C. Using the starting strain SCAU4 as a control, DNA from the caps, stems, and regenerated hyphae of overexpressing strains and the starting strain SCAU4 were extracted following the steps of the fungal DNA extraction kit. Whether the overexpressing strains still contain the hygromycin marker gene was verified by PCR.

### 2.6. Comparison of Mycelial Growth Rate and Fruiting Body Yield

The SCAU4 and *TYR7523* overexpressing strains were inoculated into regular PDA medium for 7 days. The 0.5 cm diameter hyphae punched with a punch were transferred to the parental medium, the original culture medium, and the cultivated culture medium, and incubated at a constant temperature of 30 °C. The diameter of the hyphae on the PDA plate was measured using the cross-validation method, and the growth rate (mm/d) was calculated. One-way ANOVA was performed using SPSS24.0, with 3 replicates for each strain. In addition, the cultivation packages of the SCAU4 and *TYR7523* overexpressing strains were covered with soil for fruiting, and their primordium formation time and fruiting yield were recorded.

### 2.7. Determination of Browning Degree and Tyrosinase Activity of the Starting Strain and Overexpressing Strain of Macrocybe gigantea

The fruiting bodies of SCAU4 and *TYR7523* overexpression strains with consistent size, no pest infestation, and no damage were randomly selected and sealed in PE preservation bags, and stored in a 4 °C incubator for 30 days. During this period, samples were taken every 6 days for testing, with 3 replicates per sample.

Following the method of Wu Ying [2], The CR-410 fully automatic color difference meter was used to determine the whiteness value of the fruiting body section. The color change is represented by brightness, red/green (+red, −green), yellow/blue (+yellow, −blue), and the letters are L*, A*, b*, respectively. L* also indicates the whiteness value. The larger the value, the fresher the giant mushroom, the lower the L*, and the more serious the browning degree of the fruiting bodies. Determination method: three points as tissue section, measure the L*, A*, and b* values, and set three sets of replicates for each treatment.

Referring to the method of Yu Xiaoming et al. [10], a spectrophotometer was used to measure the changes in absorbance values of each sample of *M. gigantea* at 480 nm within 3 min. Enzyme activity units were expressed as 0.001 changes per minute, and the results were expressed as U/g FW.

### 2.8. Measurement of Other Enzyme Activity Indicators

The activity of polyphenol oxidase (PPO) was measured following the method outlined by Yu Xiaoming [10] The activity of laccase (LAC) was determined using the method described by Li Ping et al. [25]. The activity of phenylalanine ammonia-lyase (PAL) was assayed according to the procedure provided by Seda Şirin [26]. For the determination of total phenolic content, the method referenced was that of Şirin [27].

## 3. Results and Analysis

### 3.1. Results of the Cloning Experiments of the gpd Promoter and the TYR7523 Gene

PCR amplification was performed using the DNA and cDNA of the *M. gigantea* SCAU4 as templates, resulting in a 1019 bp gpd promoter and a 381 bp *TYR7523* gene (Figure 1A,B), indicating the preliminary successful cloning of the two fragments. After gel cutting and TA cloning, the linked product was transferred into DH5 α—competent cells. Specific bands were found at positions 380 bp and 1000 bp by bacterial liquid PCR, further indicating the successful acquisition of the target gene (Figure 1C).

### 3.2. Construction of Overexpression Vector

PCR amplification was performed using plasmid pCAMBIA1303-TrpC Hygro-gpdA GFP and linker products pMD gpd and pMD *TYR7523* as templates, NOS terminator fragments with enzyme cleavage sites (Figure 2A), gpd promoter, and *TYR7523* gene were obtained. Then, the linearized vector plastid4-ATMT was ligated with the amplified target sequence using homologous recombinase. The obtained recombinant plasmid was transferred into DH5 α—competent cells by heat shock method. Positive single colonies were selected and shaken for PCR validation. After successful validation, the plasmid was sent to the biological organism for sequencing. The plasmid with correct sequencing was named *plastid4-GT7523* (Figure 2B). At the same time, the correct plasmid was subjected to Apa Ι and Sac Ι double enzyme digestion validation, as shown in Figure 2C, resulting in a linear vector of 8.5 bp in size and a *gpd-TYR7523* Nos fragment of 1.7 kb, indicating the successful construction of the plasmid *4-GT7523* vector.

### 3.3. Results of the Agrobacterium-Mediated Genetic Transformation Experiments

The single colonies grown on resistant plates were randomly selected and expanded for 24 h. An appropriate amount of bacterial solution was taken for PCR verification of the presence of the Hyg marker gene. The results showed that a fragment of the desired size appeared around 965 bp (Figure 3A), indicating that plasmid *plasmid4-GT7523* is transferred into *Agrobacterium*. Subsequently, *Agrobacterium tumefaciens* containing *plasmid 4-GT7523* was used to infect the mycelium of the *M. gigantea* SCAU4. After co-culture (Figure 3B) and screening culture (Figure 3C), a positive transformed strain was obtained.

### 3.4. Identification of Positive Transformants

After screening, six positive transformants with resistance to hygromycin were obtained. The DNA of these six positive transformants was extracted for validation of the Hyg gene, and a target band of about 965 bp was obtained for all of them (Figure 4A). It preliminarily proved that the vector containing the Hyg marker gene was integrated into the SCAU4 genome of the *M. gigantea*. Afterward, a positive transformant and the starting strain SCAU4 were cultured at 30 °C, and the total RNA was extracted and reverse transcribed into cDNA for qPCR verification. The results showed that the expression level of TYR7523 in the overexpressing strain was 3.47 times higher than that in the starting strain SCAU4 (Figure 4B), further indicating that the *TYR7523* gene was overexpressed.

The caps, stems, and regenerated hyphae of the *TYR7523* overexpression strain and the starting strain SCAU4 fruiting body were collected and extracted their DNA, and they were used as templates to amplify the Hyg gene. The results showed that all three samples of the *TYR7523* overexpression strain fruiting body had a band at 965 bp (Figure 4C), while the starting strain SCAU4 fruiting body had no band, indicating that a stable genetic overexpression strain was obtained.

### 3.5. Results of Mycelial Growth Rate Comparing with Fruiting Body Yield

The SCAU4 strain and *OE7523* strain were inoculated on the mother culture medium, original culture medium, and cultivated culture medium for cultivation, and their growth rates on the three media were compared to see whether there were any differences. From Figure 5 it can be seen that the growth rates of SCAU4 strain and *OE7523* strain were similar in the parent culture medium, and there was no significant difference. However, for the original culture medium and cultivated culture medium, the growth rate of *OE7523* strain was significantly different from that of SCAU4 strain. Furthermore, in the mushroom growth experiment, the fruiting body primordia of the *OE7523* strain was found to form at 7 days after covering the soil, while the SCAU 4 strain did not begin to form until 10. In the later harvesting process, it was also found that the mushroom yield of the *OE7523* strain was higher than that of the SCAU4 strain, suggesting that the *TYR7523* gene may be related to the formation of the fruiting body primordia of the *M. gigantea*.

### 3.6. Determination of Browning Degree and Tyrosinase Activity

The changes in the browning degree of the fruiting bodies of the overexpression strain *OE7523* during storage were shown in Figure 6A, using the *M. gigantea* SCAU4 strain as a control. The browning degree of overexpression strain *OE7523* showed a relatively gentle upward trend before the 12th days after harvesting, followed by a significant increase thereafter, while the browning degree of SCAU4 strain showed a relatively gentle increase throughout the storage period, indicating that the SCAU4 strain of the *M. gigantea* had less obvious browning within 30 days after harvesting. There was no significant difference between the two strains before 12 days postharvest, but there was a significant difference after 18 days postharvest (*p* < 0.05), indicating that overexpression of the *TYR7523* gene exacerbated browning of the fruiting body of the *M. gigantea*.

The comparison between the changes in tyrosinase activity of the fruiting body of overexpression strain *OE7523* and the parent strain SCAU4 during storage was shown in Figure 6B. During storage, the tyrosinase activity of both strains showed a wave-like trend of first decreasing, increasing, then decreasing, and then increasing, indicating that harvesting induced the occurrence of tyrosinase-dominated browning reaction, consumed tyrosinase, and decreased activity, but also activated the synthesis pathway of tyrosinase, leading to an increase in activity. The tyrosinase activity of the fruiting body of strain *OE7523* was significantly higher than that of the SCAU4 maternal strain from the 6th days after harvesting (*p* < 0.05), indicating that overexpression of the *TYR7523* gene exacerbated the accumulation of tyrosinase in the fruiting body, consistent with the results of browning index.

### 3.7. Polyphenol Oxidase (PPO) Activity

Using the *Macrocybe gigantea* Massee SCAU4 strain as a control, the changes in PPO activity during the storage period of the harvested fruiting bodies of the overexpressed *OE7523* strain are shown in Figure 7A. During the storage period, the overall PPO activity of both strains showed a wavy pattern, with peaks occurring on the 12th day. Starting from the 18th day, there were significant differences in PPO activity between the *OE7523* strain and the SCAU4 strain, with the PPO activity of the *OE7523* strain being significantly higher than that of the SCAU4 strain.

### 3.8. Laccase (LAC) Activity

Using the *Macrocybe gigantea* Massee SCAU4 strain as a control, the changes in LAC activity during the storage period of the harvested fruiting bodies of the overexpressed *OE7523* strain are shown in Figure 7B. Throughout the storage period, the LAC activity of the *OE7523* strain showed a trend of rapidly decreasing shortly after harvest, followed by an increase and then a continuous decline. In contrast, the LAC activity of the SCAU4 strain showed a trend of slowly decreasing initially, followed by a fluctuation with an increase on the 30th day.

### 3.9. Total Phenolic Content

Using the *Macrocybe gigantea* Massee SCAU4 strain as a control, Figure 7C illustrates the changes in total phenolic content during post-harvest storage of the overexpressing *OE7523* strain’s fruiting bodies. Throughout the storage period, the total phenolic content of both strains exhibited a wavy pattern of increasing, decreasing, increasing again, and then decreasing once more. In the early storage stages, there were no significant differences between the two strains. However, significant differences in total phenolic content emerged between the two strains in the later storage stages, which was consistent with the progression of browning and changes in enzyme activity. As the post-harvest storage duration extended, it promoted the occurrence of browning reactions.

### 3.10. TYR7523 Gene Expression Analysis

Figure 8A: Comparison of *TYR7523* expression during different developmental stages of the fruiting bodies of SCAU4 and *OE7523* strains. The results indicate that this gene is expressed at all developmental stages. Throughout the entire developmental period, the expression level in the *OE7523* was significantly higher than that in the SCAU4 strain, particularly during the flattened and curled stages, where *TYR7523* showed a significant upregulation of up to 4.33 times. This confirms that not only does the overexpressing vector *plasmid4-GT7523* fully express in the mycelium of Agaricus giganteus, but it also achieves overexpression in the fruiting bodies during their cultivation and development.

Figure 8B: Expression of *TYR7523* in mature (hemispherical stage) fruiting bodies of SCAU4 and *OE7523* strains during different post-harvest storage periods. Across the six storage periods, the expression level of *TYR7523* in the *OE7523* strain was significantly higher than that in the SCAU4 strain. On the 0th day of storage, the fold difference was the highest, with an upregulation of 3.35 times in the *OE7523* strain. As the storage time increased, the difference in upregulation gradually narrowed time.

## 4. Discussion and Conclusions

Previous studies have found that no-browning in the fruiting bodies of *Macrocybe gigantea* Masseewas is related to the low expression of tyrosinase genes in their fruiting bodies. Therefore, the function of the tyrosine gene *TYR7523* was preliminarily explored using overexpression methods in this study. Gene overexpression is mainly achieved by constructing overexpression vectors containing strong promoters, enhancers, and other elements to overexpress the target gene in transformed strains [28]. In this study, the dual vector plasmd4 ATMT was used as the expression vector to overexpress the *TYR7523* gene. The *TYR7523* gene was then transferred into the SCAU4 hyphae of the *M. gigantea* through *Agrobacterium*-mediated genetic transformation. After passage culture, the overexpression strain *OE7523* was obtained. The browning degree and tyrosinase activity of the *OE7523* strain were significantly higher than those of the SCAU4 strain in the later stage of storage, which is similar to previous research results. Sato et al. [29]. Transformed the tyrosinase gene overexpression vector PChG gTs into the shiitake mushroom strain and found that the tyrosinase activity was also significantly increased. Gao et al. [30]. Overexpressed the gene mel C encoding tyrosinase and also found that the melanin content was significantly increased. In addition, it was found that the time required for the formation of the *OE7523* strain primordia was shorter than that of the starting strain SCAU4, suggesting that the expression level of the *TYR7523* gene may have an impact on the formation stage of the fruiting body primordia [31], but further research is still needed to confirm this.

Tyrosinase is a key enzyme in melanin synthesis, which can catalyze the oxidation of tyrosine to dopa, which forms dopa quinone and then control the activity of melanocytes and affect the quality of fruits and vegetables during the storage period. During the storage period, the activity of *TYR* decreased, rose, decreased, and rose again. Picking and storage induced the browning reaction dominated by tyrosinase and the activity decreased, but at the same time, the activity of the enzyme, and the activity gradually decreased [32]. From the 6th day of post-harvest storage, the *TYR* activity of *OE7523* strain was significantly higher than the TYR activity of SCAU 4 strain (*p* < 0.05), indicating that the overexpression of tyrosinase gene *TYR7523* leads to increased tyrosinase activity in post-harvest cells. TYR activity in the SCAU 4 strain was overall maintained at low levels and showed little difference in activity changes across the time periods. However, the TYR activity of the *OE7523* strain was very different from the SCAU 4 strain at the 18 d of harvest, consistent with the significance of the browning index, indicating that the overexpression of the tyrosinase gene *TYR7523* improved the tyrosinase activity in the fruiting body and accelerated the browning of the fruiting body of the overexpressing strain.

Polyphenol oxidase (PPO) in edible mushroom fruiting bodies can oxidize phenolic compounds into quinones, which then polymerize to form brown substances, leading to discoloration of the fruiting body tissue Liu Qin et al. [33]. In this study, the PPO activity of the *OE7523* strain was significantly higher than that of the SCAU 4 strain, aligning with the findings of Liu Chunyan et al. [34], who reported higher polyphenol oxidase activity in susceptible strains of Agaricus bisporus prone to browning. The change in PPO activity in the *OE7523* strain corresponds with its tyrosine (TYR) activity and browning index results, indicating that when PPO activity is low in Volvariella gigantea fruiting bodies, the enzymatic browning reaction progresses slowly, better preserving the quality of the fruiting bodies. Conversely, higher PPO activity promotes a higher degree of oxidation of phenolic compounds in the fruiting bodies, resulting in a higher browning index.

Laccase can catalyze the oxidation of various phenolic and non-phenolic substances, ultimately leading to the oxidation or decomposition of the substrate, which affects the nutritional value and quality of edible mushrooms during storage in Gong Rui [35]. Throughout the storage period, the trends in laccase (LAC) activity between the two strains differed significantly. There were significant differences in LAC activity trends between the two strains during the early and late storage periods, while no significant difference was observed during the middle storage period. This may be due to changes in laccase isoenzymes in the *OE7523* strain caused by the overexpression vector, resulting in differences in the catalytic oxidation and decomposition reactions. Similar results have been reported in studies on laccase in white-rot fungi Li Wenjuan [36]. The trends in laccase activity of the two strains did not correspond with the browning index trends, indirectly confirming that laccase is not the key enzyme affecting browning in Volvariella gigantea. Total phenols are important substrates for enzymatic browning reactions and possess strong antioxidant capacity. Their content is closely related to the degree of browning in mushrooms [37]. As post-harvest storage time increases, the browning reaction is promoted, and to meet the demand for phenolic substrates in the browning reaction, more phenolic compounds are produced in the fruiting body cells of *Macrocybe gigantea* Massee. The increase in phenolic compounds can participate in tissue healing processes. With prolonged storage, membrane lipid peroxidation intensifies, gradually damaging the cell membrane, allowing previously isolated phenolic substrates to come into contact with enzymes and react, resulting in a gradual decrease in total phenol content. This trend is consistent with the changes in total phenol content observed in post-harvest storage of *Hericium erinaceus* and *Lentinus edodes* [38,39].

In contrast, giant mushroom overexpression *OE7523* strain and SCAU 4 strain of *TYR7523* expression, detected by qRT-PCR in different development and different storage periods of fruiting body tyrosinase gene *TYR7523* expression, two strains of *TYR7523* gene expression of the fruiting body with the development of fruiting body, namely the probasal expression are the highest, speculated that *TYR7523* gene expression and the giant mushroom fruiting body formation, The expression of *TYR7523* genes was low during the hemispheric phase of the SCAU 4 strain, the optimal period of picking. The overexpression of the tyrosinase gene *TYR7523* was achieved in the real body, and the overexpression of the tyrosinase contributed to its browning enhancement. Using 0 d as a control, the expression of *TYR7523* in both strains was gradually increasing, reaching the peak at 30 d, indicating that the expression of *TYR7523* had a significant impact on the storage time of mature fruiting bodies. The storage resistance of giant mushrooms was caused by the low expression of tyrosinase.

Based on the storage characteristics of *Macrocybe gigantea*, the overexpression vector was constructed to further illustrate the connection between the low expression of tyrosinase gene *TYR7523* and the storage tolerance of giant mushrooms, which provided a reference for the further study of the mechanism of brown resistance and the development of biological engineering of edible fungi.

## 5. Convey Thanks

During the completion of this research project, we have received valuable support and help in many aspects, with deep thanks here.

First of all, I would like to thank Professor Mo Meihua for his careful guidance and selfless dedication. From the discovery of the giant mushroom and its storage characteristics, research topics, and experiment design to the article writing, every step embodies the wisdom and painstaking efforts of the tutor. My tutor’s rigorous scientific research attitude, profound academic attainments, and unremitting pursuit of scientific exploration have had a profound influence on me, which is the beacon on my academic road.

Secondly, we especially thank the School of Food Science for its experimental platform and technical support that enabled the smooth conduct of this study. I would like to thank Professor Wang Jie and Professor Xu Xuefeng for their professional guidance and valuable suggestions in experimental operation and data analysis, which greatly promoted the depth of research and the optimization of results.

In addition, this study obtained the natural science fund of Guangdong province, 2024 provincial rural revitalization strategy of special funds, Maoming city of Guangdong province Gaozhou Zhenjiang town rural science and technology correspondent team project, for experimental materials, equipment procurement, and travel research provides the necessary financial support, our sincere gratitude.

Here, I would also like to thank Tian Yi, Xiao Yuling, Wang Xueting, Yan Yamin, Xue Jianbang, and other laboratory members for their mutual support and cooperation in their daily work, and jointly creating a positive scientific research atmosphere. Their wisdom collision and unremitting efforts have laid a solid foundation for the smooth completion of the project.

Finally, I would like to thank my family and friends, whose understanding, encouragement, and company are the motivation. When I was faced with challenges and pressure, it was their silent support that enabled me to keep a positive and optimistic attitude and focus on scientific research.

Once again, I would like to express my deepest thanks to all the people who gave them help and support in the research process!

## Figures and Tables

**Figure 1 jof-11-00216-f001:**
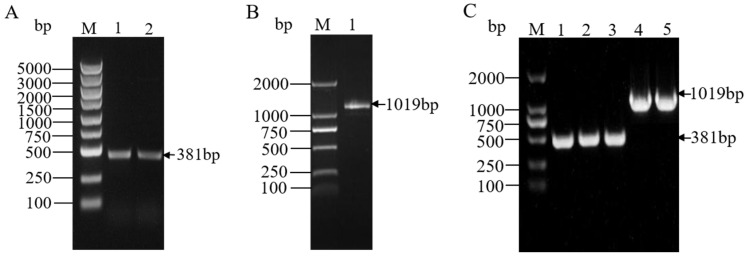
Cloning of objective genes. M: DNA marker; (**A**): *TYR7523* gene amplification results; (**B**): gpd promoter amplification results; (**C**): Bacterial liquid PCR results, 1–3: *TYR7523* gene, 4–5: gpd promoter.

**Figure 2 jof-11-00216-f002:**
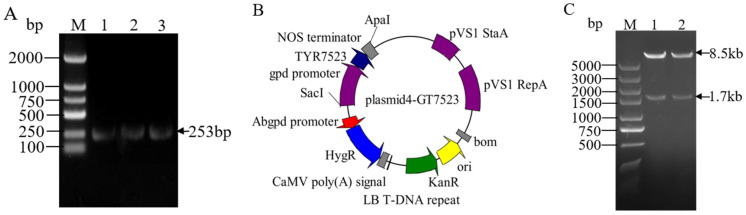
Construction of overexpression vector *plasmid4-GT7523*. M: molecular marker; (**A**): NOS terminator amplification results; (**B**): overexpression vector mapping; (**C**): restriction enzyme digestion results, 1–3: Digestion product of *plasmid4-GT7523*.

**Figure 3 jof-11-00216-f003:**
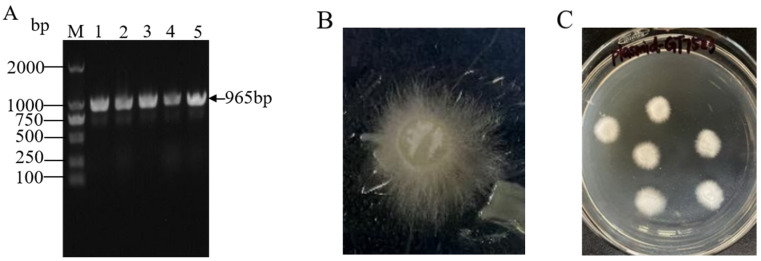
Screening and cultivation of transformants. M: DNA marker; (**A**): PCR screening results of transformants, 1–5: Hyg gene; (**B**): co-culture map; (**C**): growth map of transformants on PDA screening medium.

**Figure 4 jof-11-00216-f004:**
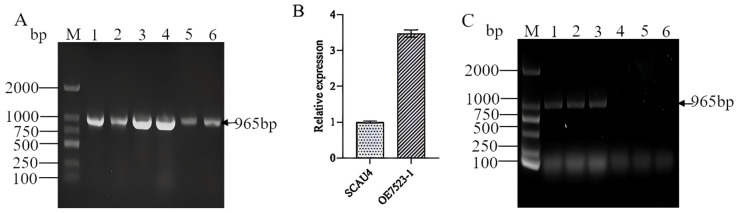
Identification of positive transformants. M: DNA marker; (**A**): PCR identification of hygromycin; (**B**): qRT-PCR verification of transformants; (**C**): PCR identification results of fruiting bodies, 1–3: amplification results of cap, stalk and rejuvenated hyphae of fruiting bodies of overexpressed strains, 4–6: Amplification results of cap, stalk and rejuvenated hyphae of fruiting bodies of original strains.

**Figure 5 jof-11-00216-f005:**
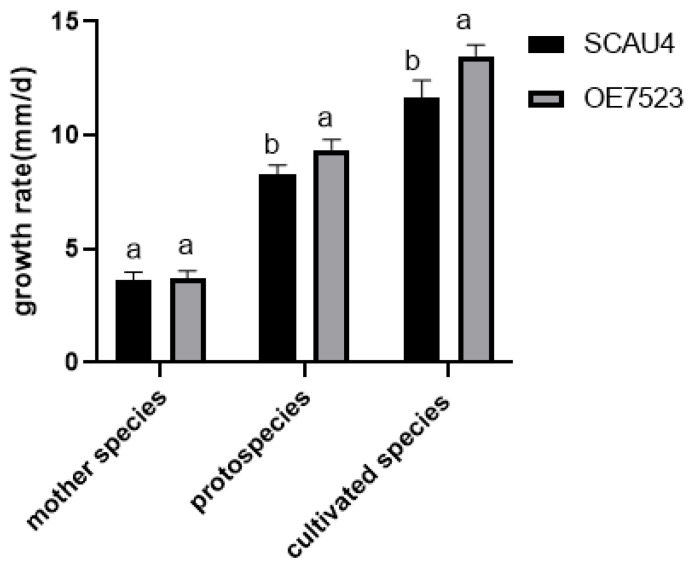
Growth rate of transformants. The mother species is the plate growth strain, the protospecies is the wheat growth strain, and cultivated species are strains grown in mycelium culture medium. Note: Letters and a indicate the difference between two strains during the same storage period, b indicates *p* > 0.05, the difference is not significant; a indicates the significant difference at the *p >* 0.01 level.

**Figure 6 jof-11-00216-f006:**
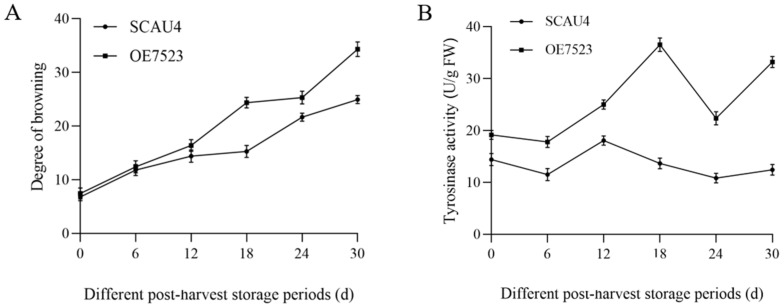
Determination of browning-related indexes. (**A**): Changes in browning degree of fruiting bodies of the starting strain SCAU4 and the overexpressed strain *OE7523* during storage; (**B**): Changes in tyrosinase activity of fruiting bodies of the starting strain SCAU4 and overexpressed strain *OE7523* during storage.

**Figure 7 jof-11-00216-f007:**
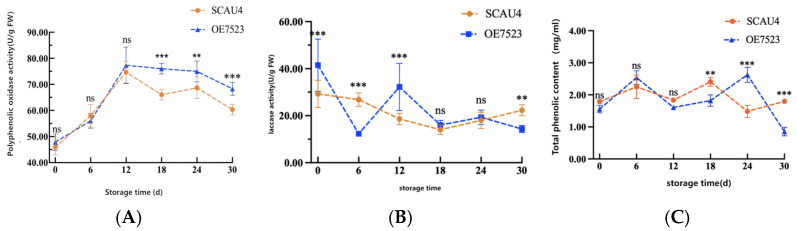
Other Enzyme Activity Indicators. (**A**): PPO Activity; (**B**): LAC Activity; (**C**): Total Phenolic Content. Note: Letters and asterisk indicate the difference between two strains during the same storage period, ns indicates *p* > 0.05, the difference is not significant; ** indicates *p >* 0.05, the difference is significant; *** indicates the significant difference at the *p* > 0.01 level.

**Figure 8 jof-11-00216-f008:**
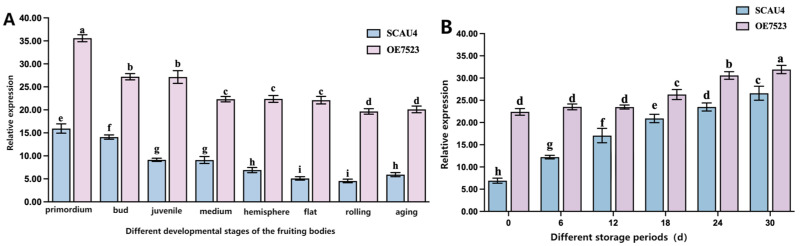
SCAU 4 and OE7523 strains (**A**): Diagram of SCAU 4 and OE7523 expression in different storage stages of fruiting bodies. (**B**): Plot of SCAU 4 and OE7523 expression at different developmental stages of the daughters. Note: letters a–i, respectively, represent the degree of significant difference (*p* < 0.05).

**Table 1 jof-11-00216-t001:** Primer design.

Primer Name	Primer Sequence (5′→3′)	Illustrate
*TYR7523-f*	CCTCCTCGAAGAACGAAAGTC	Cloning of TYR7523 gene
*TYR7523-r*	GTGCTGTTTTTCACGTTGAGGATTCC	
Gpd-f	CAGATATCTTCGGGGCCTG	Clone gpd promoter
Gpd-r	CGTGAGGGCATTTCGAAG	
M13-f	CAGGAAACAGCTATGACC	Bacterial liquid PCR validation
M13-r	TGTAAA ACGACG GCCAGT	
Insert gpd-f	CATGATTACGAATTCGAGCTCCAGATATCTTCGGGGCCTGG	Clone insertion fragment gpd
Insert gpd-r	CATCGTGAGGGCATTTCGAAGG	
*Insert 7523-f*	TTCGAAATGCCCTCACGATGGTGCAAGTCGCCAATACA	Clone Insert Fragment *TYR7523*
*Insert7523-r*	GAACGATCCTAGATGTACTCGTAACAAAGAGGGC	
Insert Nos-f	GAGTACATCTAGGATCGTTCAAACATTTGGCAATAA	Clone Insert Fragment NOS
Insert Nos-r	CAGGTCGACAGATCTGGGCCCGATCTAGTAACATAGATGACACCGCG	
Hyg-f	GGTTTCCACTATCGGCGAG	Validation of hygromycin
Hyg-r	GTCTGTCGAGAAGTTTCTGATCG	
β-tubulin-f	ACTGTCGTTGAGCCCTACAA	Amplification of internal reference genes β-tubulin
β-tubulin-r	CAAGCAAGTCGTGATACCCG	
*TYR7523-qf*	TTGAGCAGTTTCGCACAGAG	qRT-PCR analysis of *TYR7523*
*TYR7523-qr*	GGTTCAATGGTGCTACGACC	

Note: The lowercase letters in the primer sequence indicate the homologous arm sequence when constructing the vector.

## Data Availability

The original contributions presented in this study are included in the article. Further inquiries can be directed to the corresponding author.

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
