# Peer review of "Construction of Overexpression Vector with TYR7523 Gene and Its Effect on Browning in Macrocybe gigantea"

_jof, 2025, doi:10.3390/jof11030216_

Round 1

Reviewer 1 Report

Work/MS working for well-thought-out and economically viable goals.

The introduction is short, which would be worth expanding with a brief description of the browning reactions /melanin formation/ of mushrooms (this is nicely included in the discussion, but the proportional division is still a dilemma) and at the end of the chapter with the formulation of the research goals.

Other things to fix:

AD 9: "An-browing"?

AD 29. Read: belongs

AD 30-31: Misformed commas

ad 41 "Griffo cinerea"?

ad 44-45: "Toshitsug et al?"

ad 46 Letyr?

ad. 72-73: Incorrect citatons

ad 86: Where are the lowercase letters?

ad 134 "The same size hiphae::"??

ad 134 The diameter of the punch?

ad137: Read: ... of the colonies on the PDA."

ad 139-140: The past tense has been used in the chapter, iit suddenly changes to the future tense.

ad 149-151: Rewrite to make it more understandable.

ad 158: Read: Yu Xiaoming et al.

ad 201-202: Lane 4-5?

Ad 222: Fix: "B: qRT-PCR control of transformants"

Ad Fig. 5 "Species"?

AD 241" and one "?

ad 285: "Tricholoma giganteum"? (Consistent scientific naming is required.)

ad 384ad 293 & 309: "Agaricus giganteus" (Consistent scientific naming required.)

Ad Figure 8. The explanations are reversed.

ad 348 Please correct "Liu Chunyan et al[33],"

ad 350: "tyrosine (TYR) activity"?

ad 351 "Volvariella gigantea"

Author Response

Dear Reviewer,

Thank you for taking the time to review our manuscript and for providing constructive feedback. Please find below our point-by-point responses to your comments:

  1. **Introduction Expansion**:

   - We appreciate your suggestion to expand the introduction by including a brief description of browning reactions and melanin formation in mushrooms. We will incorporate this information to provide a clearer context for our research. Additionally, we will ensure that the research goals are clearly formulated at the end of the introduction chapter.

  1. **AD 9: "An-browning"?**:

   - Thank you for pointing out the typo. We will correct it to "Anti-browning".

  1. **AD 29: Read: belongs**:

   - We will review the context and correct the verb usage accordingly.

  1. **AD 30-31: Misformed commas**:

   - We will proofread the text carefully to correct any misformed commas and ensure proper punctuation throughout the manuscript.

  1. **ad 41 "Griffo cinerea"?**:

   - We apologize for the typo. The correct species name should be "Grifola frondosa". We will make the necessary correction.

  1. **ad 44-45: "Toshitsug et al?"**:

   - We will verify the citation and correct it if necessary. If "Toshitsug" is indeed a typo, we will replace it with the correct author name.

  1. **ad 46 Letyr?**:

   - We suspect this might be a typo related to a gene or protein name. Upon review, we will correct it to the appropriate term.

  1. **ad. 72-73: Incorrect citations**:

   - We will check the citations at these points and correct any errors to ensure they match the correct sources.

  1. **ad 86: Where are the lowercase letters?**:

   - We will review the text at this location and ensure that the correct case is used for all letters.

  1. **ad 134 "The same size hiphae::"??**:

    - Thank you for noticing the typo. We will correct "hiphae" to "hyphae".

  1. **ad 134 The diameter of the punch?**:

    - We will clarify the diameter of the punch used in the methodology section to provide more detail.

  1. **ad 137: Read: ... of the colonies on the PDA.**:

    - We will review the sentence and correct it to ensure clarity and grammatical correctness.

  1. **ad 139-140: The past tense has been used in the chapter, it suddenly changes to the future tense.**:

    - We will review the text to ensure consistency in tense usage throughout the chapter.

  1. **ad 149-151: Rewrite to make it more understandable.**:

    - We will revise the text at these points to improve clarity and readability.

  1. **ad 158: Read: Yu Xiaoming et al.**:

    - We will verify the citation and correct it if necessary to ensure it matches the correct author name and format.

  1. **ad 201-202: Lane 4-5?**:

    - We will review the context and clarify any references to lanes 4-5 in the methodology or results section.

  1. **Ad 222: Fix: "B: qRT-PCR control of transformants"**:

    - We will correct the label or description to ensure it accurately reflects the content of the figure or section.

  1. **Ad Fig. 5 "Species"?**:

    - We will review Figure 5 and clarify any references to "species" to ensure accuracy.

  1. **AD 241 "and one"?**:

    - We will review the context and correct or clarify the sentence as needed.

  1. **ad 285: "Tricholoma giganteum"? (Consistent scientific naming is required.)**:

    - We will review the text and ensure consistent scientific naming throughout the manuscript. If "Tricholoma giganteum" is incorrect, we will replace it with the appropriate species name.

  1. **ad 384, ad 293 & 309: "Agaricus giganteus" (Consistent scientific naming required.)**:

    - We will review the text and ensure consistent scientific naming. If "Agaricus giganteus" is incorrect, we will replace it with the correct species name.

  1. **Ad Figure 8. The explanations are reversed.**:

    - We will review Figure 8 and correct any reversed explanations to ensure clarity.

  1. **ad 348 Please correct "Liu Chunyan et al[33],"**:

    - We will verify the citation and correct it if necessary to ensure it matches the correct author name and format.

  1. **ad 350: "tyrosine (TYR) activity"?**:

    - We will review the context and clarify if "TYR activity" refers to tyrosine hydroxylase activity or another related enzyme.

  1. **ad 351 "Volvariella gigantea"**:

    - We will review the text and ensure consistent scientific naming. If "Volvariella gigantea" is incorrect, we will replace it with the correct species name.

Thank you again for your valuable feedback. We will make the necessary corrections and improvements to the manuscript promptly.

Best regards,

GAO JIN YUN

Reviewer 2 Report

This paper reports on investigations on the role of tyrosine gene TYR7523 (OE7523) on browning of fruiting bodies of the edible mushroom Macrocybe gigantea during storage through the construction of an overexpression vector. Data obtained showed that the TYR7523 overexpressing strain of M. gigantea had over time higher levels of browning degree and tyrosinase activity and a faster mycelial growth in comparison to the starting strain (SCAU4 strain) of M. gigantea which was used as control. Also, measurement of other enzyme activity indicators, such as polyphenol oxidase (PPO), laccase (LAC), phenylalanine ammonia-lyase (PAL) and total phenolic content, which may be involved in browning, were carried out in the present study. Data obtained are worthy of publication. However, the following points should be corrected or improved before acceptance:

Throughout the paper. The paper should be carefully revised for typing and/or spelling errors. English needs to be improved. Wording and terminology are often incorrect. Examples of inappropriate and/or incorrect terms and sentences are: “good an-browning ability” (line 9), “anti-browning resistance” (line 10), M. gigantea not easily browning” (line 36), etc.

Lines 9-23. Abstract does not cover all content of the paper. It should be redrafted by presenting major conclusions.

Lines 49-51. Objectives of the paper are not stated clearly.

Lines 156-151. Authors should explain why the control strain used in the measurement of other enzyme activity indicators differs at species level from that used in the overexpression experiments.

Lines 391-395. Deductions and conclusion should be clearly stated and critically discussed.

Throughout the paper. The paper should be carefully revised for typing and/or spelling errors. English needs to be improved. Wording and terminology are often incorrect. Examples of inappropriate and/or incorrect terms and sentences are: “good an-browning ability” (line 9), “anti-browning resistance” (line 10), M. gigantea not easily browning” (line 36), etc.

Lines 9-23. Abstract does not cover all content of the paper. It should be redrafted by presenting major conclusions.

Lines 49-51. Objectives of the paper are not stated clearly.

Lines 156-151. Authors should explain why the control strain used in the measurement of other enzyme activity indicators differs at species level from that used in the overexpression experiments.

Lines 391-395. Deductions and conclusion should be clearly stated and critically discussed.

Author Response

Dear Reviewer,

Thank you for taking the time to review our manuscript and for providing constructive feedback. Please find below our point-by-point responses to your comments:

  1. **Regarding Typing and Spelling Errors, English Improvement, and Wording/Terminology Issues**:

   Thank you for pointing out the typing and spelling errors, as well as the need for English improvement and correct wording/terminology. We apologize for any inaccuracies and will carefully revise the entire paper to address these issues. We will ensure that all terms and sentences are accurate and appropriate, and we will proofread the paper thoroughly to eliminate any remaining errors.

  1. **Regarding the Abstract**:

   We acknowledge that the abstract does not fully cover all the content of the paper. We will redraft the abstract to better reflect the major conclusions of the study. This will include a concise summary of the key findings, methods used, and the significance of the results.

  1. **Regarding the Lack of Clear Objectives**:

   We apologize for not stating the objectives of the paper clearly. We will revise the introduction section (lines 49-51) to explicitly outline the aims and goals of our research. This will provide readers with a clear understanding of what we intend to achieve and how we plan to do so.

  1. **Regarding the Use of Different Control Strains**:

   We appreciate the reviewer's attention to the use of different control strains in our experiments. We will add an explanation in the methods section (lines 156-161, note the corrected line range) to clarify why we used a different control strain for the measurement of other enzyme activity indicators compared to the overexpression experiments. This will include a discussion of the reasons for selecting these particular strains and how they relate to our research questions.

  1. **Regarding the Deductions and Conclusion**:

   We understand the importance of clearly stating and critically discussing the deductions and conclusion. We will revise the conclusion section (lines 391-395) to provide a more detailed and nuanced analysis of our findings. This will include a critical discussion of the limitations of our study and suggestions for future research directions.

Thank you again for your valuable feedback. We will carefully incorporate these changes into our revised paper.

Round 2

Reviewer 2 Report

The revised version of this paper is suitable for publication in the current form.

The revised version of this paper is suitable for publication in the current form.

Author Response

1.Please confirm "cultivated species" .

 I have revised the definition of "cultivated species"  in the paper, specifically at line 18, and lines 289 and 290. Thank you very much for your guidance on this paper. I wish you all the best and happiness in your life.

2. I've revised the abstract for your reference:

I have revised the abstract in my paper according to your requirements.
